



# Antarctic ice sheet paleo-constraint database

Benoit S. Lecavalier[1], Lev Tarasov[1], Greg Balco[2], Perry Spector[2], Claus-Dieter Hillenbrand[3], Christo Buizert[4], Catherine Ritz[5], Marion Leduc-Leballeur[6], Robert Mulvaney[3], Pippa L. Whitehouse[7], Michael J. Bentley[7], Jonathan Bamber[8,9]

[1]Department of Physics and Physical Oceanography, Memorial University, St. John's, Canada
[2]Berkeley Geochronology Center, Berkeley, California, USA
[3]British Antarctic Survey, Cambridge, UK
[4]College of Earth, Ocean and Atmospheric Sciences, Oregon State University, Corvallis OR, USA
[5]Université Grenoble Alpes, CNRS, IRD, IGE, Grenoble, France
[6]Institute of Applied Physics National Research Council, Florence, Italy
[7]Department of Geography, Durham University, Durham, UK
[8]Bristol Glaciology Centre, School of Geographical Sciences, University of Bristol, Bristol, UK
[9]Department of Aerospace and Geodesy, Data Science in Earth Observation, Technical University of Munich, Munich, Germany

Correspondence to: Benoit S. Lecavalier (b.lecavalier@mun.ca)

**Abstract.** We present a database of observational constraints on past Antarctic ice sheet changes during the last glacial cycle intended to consolidate the observations that represent our understanding of past Antarctic changes, for state-space estimation, and paleo-model calibrations. The database is a major expansion of the initial work of Briggs and Tarasov (2013). It includes new data types and multi-tier data quality assessment. The updated constraint database "AntICE2" consists of observations of past grounded and floating ice sheet extent, past ice thickness, past relative sea level, borehole temperature profiles, and present-day bedrock displacement rates. In addition to paleo-observations, the present-day ice sheet geometry and surface ice velocities are incorporated to constrain the present-day ice sheet configuration. The method by which the data is curated using explicitly defined criteria is detailed. Moreover, the observational uncertainties are specified. The methodology by which the constraint database can be applied to evaluate a given ice sheet reconstruction is discussed. The implementation of the "AntICE2" database for Antarctic ice sheet model calibrations will improve Antarctic ice sheet predictions during past warm and cold periods and yield more robust paleo model spin ups for forecasting future ice sheet changes.

## 1. Introduction

Numerical ice sheet models have been applied to reconstruct past continental-scale ice sheet changes in Antarctica for decades (Whitehouse et al., 2012a; Golledge et al., 2014a; Briggs et al., 2014; Huybrechts, 2002; Pollard and DeConto, 2009). However, given the host of uncertainties in such modelling, assessment of the correspondence between model results and past Antarctic Ice Sheet (AIS) evolution requires (among other things) a quality-controlled constraint database with carefully assessed observational uncertainties. To date, only one database is publicly available (Briggs and Tarasov, 2013), and it suffers from some key limitations. Specifically, many regions, such as in the ice sheet interior, lack any observational constraints, and the data quality was not explicitly evaluated and specified through standardized criteria. Paleo ice sheet modelling has a host of uncertainties associated with: initial and boundary conditions, physical processes, and their numerical representation. As such, inferences of ice sheet evolution must be meaningfully constrained against paleo and present-day (PD) data. This requires an accessible database with well-defined observational uncertainties and a clear understanding of model limitations.

The AIS has consistently been identified as a dominant source of uncertainty in predicting past and future global sea-level change (Meredith et al., 2019; Fox-Kemper et al., 2021). Previous studies have generated a wide range of future AIS projections (Little et al., 2013; Levermann et al., 2014; Ritz et al., 2015; Ruckert et al., 2017; Golledge et al., 2015; DeConto and Pollard, 2016) and paleo retrodictions (Whitehouse et al., 2012a; Golledge et al., 2014a; Briggs et al.,



2014; Argus et al., 2014; DeConto and Pollard, 2016; Huybrechts, 2002; Simms et al., 2019; Albrecht et al., 2019),
often with poorly defined confidence intervals. Most often these issues are dealt with via parametric tuning to generate "reasonable" predictions and upper/lower bound estimates (e.g. Golledge et al., 2014b; DeConto and Pollard, 2016). The integration of a constraint database would help quantify what is deemed a "reasonable" result. Additionally, most previous studies inadequately explored parametric uncertainties, did not account for structural uncertainties of the model, and only applied a small set of observational constraints. An incomplete uncertainty
assessment for model results largely nullifies the utility of the model predictions in the context of understanding the actual physical system under consideration (Tarasov and Goldstein, 2021).

In this study we provide an overview of a data-quality curated Antarctic constraint database intended to characterize the past evolution of the Antarctic ice sheet and for ice sheet models. Key features are a quality classification and
careful specification of data uncertainties. The variety of data types is presented along with spatial and temporal information. A general overview is provided that discusses the data-system relationship and observational uncertainties. In addition, we discuss the future inclusion of additional data types, such as the age structure of the ice, and highlight outstanding issues and community challenges.

## 2. AntICE2 constraints

The updated community Antarctic ICe sheet Evolution observational constraint (AntICE) database version 2 (henceforth referred to as AntICE2) builds on the initial work of Briggs and Tarasov (2013) by integrating additional data since the original publication, including new data types. The updated database comprises observations of (1) past grounded ice and ice shelf extent (paleoEXT), (2) past ice sheet thickness (paleoH), (3) past relative sea level (paleoRSL), (4) borehole temperature profiles (boreTemp), and (5) Global Positioning System (GPS) observations of
PD uplift rates (rdotGPS). Figure 1 shows a summary of the data types in the AntICE2 database and their spatial coverage. In addition to these observations, the PD ice sheet geometry (surface elevation, ice thickness, basal topography) and surface ice velocities are provided. This major revision of the AntICE database more than quintuples the direct observational constraints from 203 to 1023 (excluding the PD AIS geometry and surface velocity field). The database is open-source (https://theghub.org/resources/4884/about) and available in the online supplementary
materials. The curation of data within the AntICE2 database was based on design criteria that excluded low quality, inconsistent, and superfluous data. If the inference of past ice sheet changes are not increased when a data point is considered, then it is excluded to prevent database bloating. The curation criteria were established by the collective authorship of this study.

To calibrate or history match a model (Tarasov and Goldstein, 2021), it is necessary to compare model simulations to observations. For such comparison to have meaning, it logically follows that the relationship between each data point and the actual physical system must be specified. The selection of data with a high signal (measured quantity) to data uncertainty ratio can strongly facilitate the inference process. To calculate a data-model misfit score for a given observation, the observation must include location data (latitude, longitude), age data determined with a well-
established dating technique, and it must quantify the relationship between the proxy observation and the characteristic (i.e., the recorded change in the ice sheet) it constrains. For example, past ice thickness inferred from the elevation of an erratic boulder with an age determined by [10]Be cosmogenic nuclide exposure dating, constrains the time when ice-sheet thinning caused the ice surface to fall below the altitude of the sample. The paleo data is categorized by site, where data from nearby samples (typically within <10 km distance) are clustered together,
thereby yielding a time-series at a given site (paleoRSL, paleoH). The exact spatial coordinates of the data are taken from the source publication and transcribed into the database. The sites of the paleo data in Fig. 1 and 2 show the average location of all the data clusters near a given site.





Figure 1: Antarctic ICe sheet Evolution database version 2 (AntICE2) summary plot. The Antarctic basemap was generated using Quantarctica (Matsuoka et al., 2021).

Each site has a unique four-digit identifier (Fig. 2). The first digit represents the data-type (paleoH = 1, paleoEXT = 2, paleoRSL = 9, boreTemp = 5, rdotGPS = 8), the second digit designates the drainage basin sector (Dronning Maud-Enderby Land = 1, Lambert-Amery = 2, Wilkes-Victoria Land = 3, Ross Sea = 4, Amundsen Sea and Bellingshausen Sea = 5, Antarctic Peninsula = 6, Weddell Sea = 7; sector boundaries are shown in Fig. 2), and the last two digits identify the site within each sector (westernmost site = 1, increasing by one eastward following the coast). The types of paleo data along with full references are found in supplementary Tables S1–S5. The method by which the data is processed and interpreted is described below.


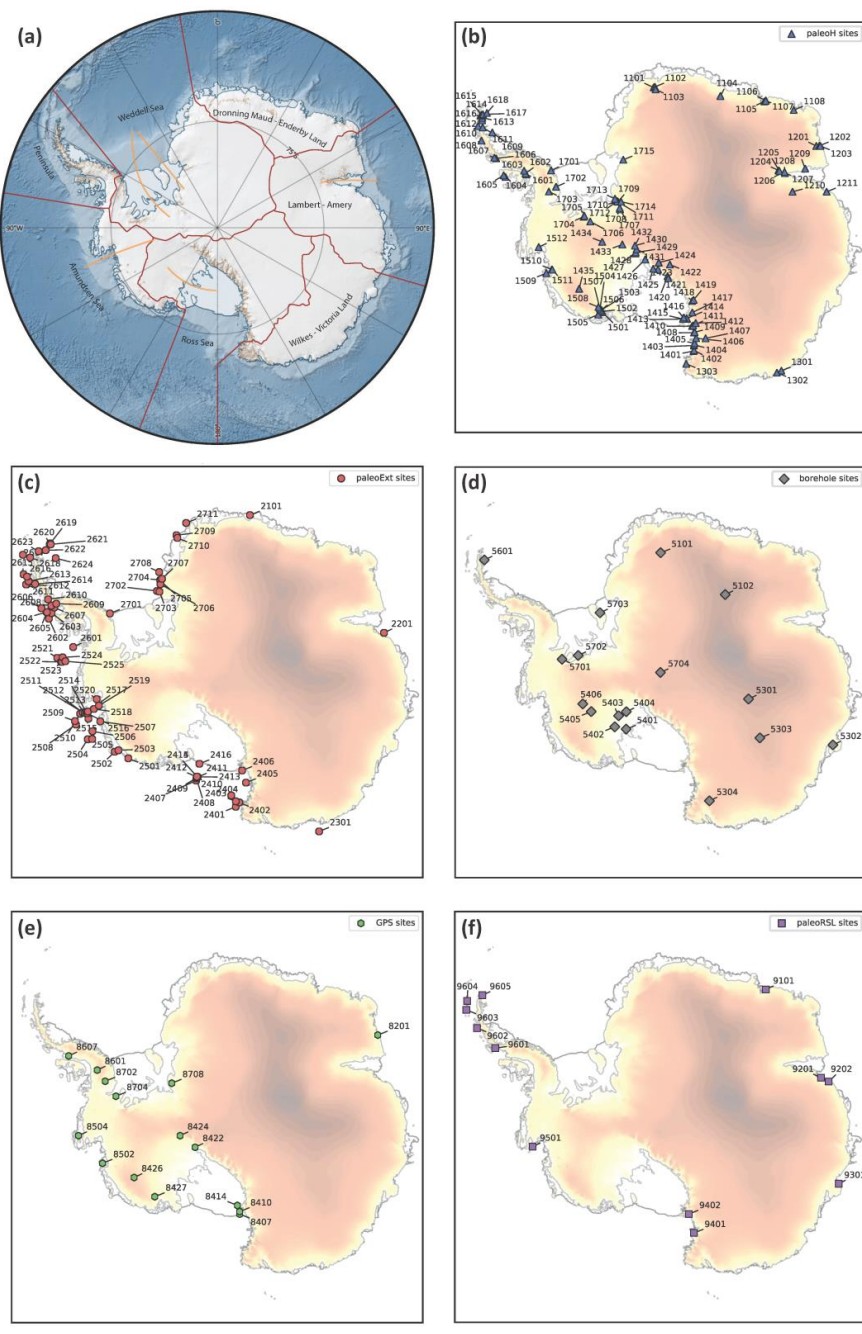

Figure 2: a) Clustered IMBIE2 ice-drainage basins (boundaries between clusters marked by red lines), key cross
section profiles (orange lines), and place names mentioned in the text; b - f) are the sites with past ice thickness data
(paleoH), past ice extent data (paleoExt), ice core borehole temperature profiles (ICbore), present-day uplift rates
(rdotGPS), and past relative sea level data (paleoRSL) respectively. The basemap shown in a) was generated using



Quantarctica (Matsuoka et al., 2021). The surface elevation shown in b-f) is based on the BedMachine Antarctica
version 2 dataset (Morlighem et al., 2020).

## 2.1 Paleo ice sheet thickness

When an ice sheet recedes and thins, entrained terrigenous detritus in the ice is deposited on newly exposed land.
The geographic coordinates, elevation, and exposure age of the bedrock/erratic sample, provides a point estimate
of the location of the ice surface/margin at the time of exposure. Note that while the measured elevation is relative
to PD sea level, the elevation at the time of initial exposure is unknown without knowledge of the GIA history.
However, the GIA estimate is not needed, if the measurement is treated as a direct constraint on past ice thickness
rather than ice surface elevation. In Antarctica, these measurements are mostly conducted along the slope of ice-
free mountains/nunataks piercing through the ice sheet surface (e.g. Balco et al., 2016; Small et al., 2019). When
many samples along a transect across a topographical slope are analysed, one can reconstruct a chronology of paleo
ice sheet thinning since the last ice thickness maximum in the region (Stone et al., 2003; Ackert et al., 2007). This is
illustrated in Fig. 3 showing sample elevation histories from different sites during the deglaciation following the Last
Glacial Maximum (LGM: ca. 19-23 ka) and in Fig. S1-S102 showing the entire AntICE2 paleoH dataset.

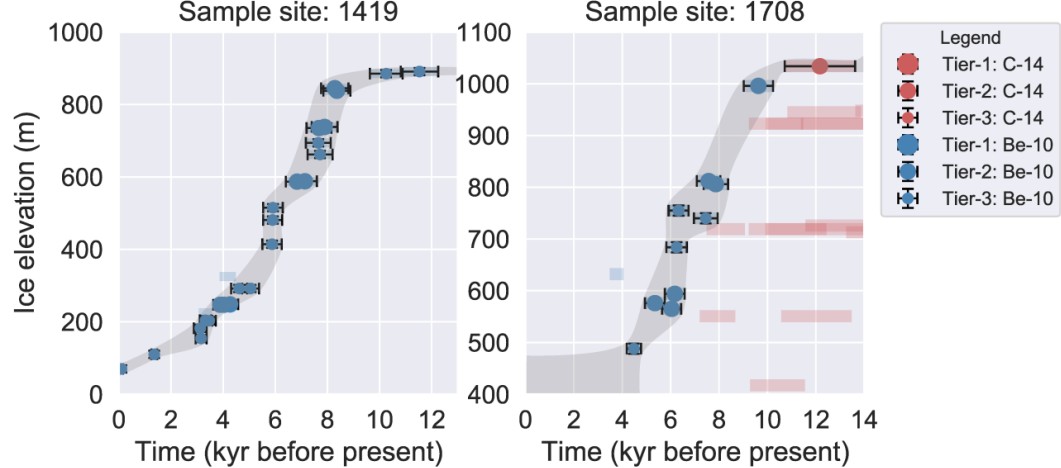

Figure 3: Sample past ice thickness (paleoH) data to illustrate the data quality and tier assignment. The elevation
data is converted to ice thickness data using the Bedmachine basal topography data. The grey band illustrates the
most probable history at the given site. The blue and red transparent bands represent other C-14 and Be-10 data
not assigned into a quality tier.

Cosmogenic-nuclide exposure dating on bedrock and erratics is the primary method used to establish the timing of
deglaciation of terrestrial sites (Bentley et al., 2006; Johnson et al., 2017; Nichols et al., 2019). The method entails
the measurement of radioactive or stable nuclides isotopic concentrations ($^{10}$Be, $^{26}$Al, $^{3}$He, $^{21}$Ne, $^{36}$Cl and $^{14}$C) which
accumulate in rock surfaces exposed to the atmosphere and therefore to the cosmic-ray flux measured for
cosmogenic nuclide exposure dating. In the case of these isotopes, the nuclide concentration builds up when a rock
exposed to the atmosphere is bombarded by cosmic rays (Ackert et al., 1999; Stone et al., 2003; Ackert et al., 2007).
Using the nuclide concentration and its radioactive half-life, the time when a rock was first exposed to cosmic rays,
i.e. its exposure age and, thus, the deglaciation age of its location, can be calculated.

The interpretation of the deglaciation age can be complicated when erratics are absent or were redeposited (e.g.,
down a mountain slope), when the dated bedrock surface has been sufficiently eroded to remove cosmogenic
nuclides accumulated during prior exposure periods, and/or when the site has subsequently been reburied by
ice/snow/sediment or shielded by topography. In a case where the cosmogenic nuclide clock was not sufficiently



reset, and thus past nuclide concentrations persist, the sample would suffer from significant inheritance of pre-ice cover exposure to cosmic rays. Given the limited number of areas in Antarctica where bedrock/erratics are exposed today, the total resulting number of collected samples is relatively low. This makes it difficult to identify when inheritance is an issue unless significant sample numbers are collected or paired $^{10}Be$-$^{26}Al$ dating is performed. For a complete description of the cosmogenic nuclide exposure dating methodology and its challenges, we refer the reader to previous studies (Ackert et al., 1999; Stone et al., 2003; Bentley et al., 2006; Mackintosh et al., 2007; Balco et al., 2016; Johnson et al., 2017).

An informal cosmogenic-nuclide exposure-age database, termed ICE-D, already exists and facilitates accessibility to raw data and derived exposure ages. The ICE-D database is inclusive and illustrates the conflicting and complex exposure histories in many regions. Quality control and processing of the data is required since many samples suffer from inheritance, and some regions provide an inconsistent record of past ice surface lowering (younger samples being higher than older samples). The deglaciation age is often inferred by the highest and youngest erratic sample

(Bentley et al., 2006) with older bedrock samples at a similar elevation being discounted. Alternatively, a mean age of several samples for a site may be calculated (Todd et al., 2010). In the original AntICE database (Briggs and Tarasov, 2013), the exposure ages and uncertainties were taken directly from the literature rather than recalibrating the ages for overall consistency, in part because the raw data were often inaccessible. The ICE-D database addresses this issue by using a single, up-to-date method to calculate all cosmogenic-nuclide exposure ages. Exposure ages used in this

compilation were calculated using the "LSDn" scaling method of Lifton et al. (2014) as implemented in version 3 of the online exposure age calculator described by Balco et al. (2008) and subsequently updated Balco (2020). Production rate calibration for $^3He$ in pyroxene and olivine, $^{10}Be$ in quartz, and $^{26}Al$ in quartz uses the "primary" calibration data sets of Borchers et al. (2016). Production rate calibration for in-situ $^{14}C$ is based on measurements of the CRONUS-A quartz standard and the assumption that the concentration in this sample is at production-decay

saturation, as described in Nichols et al. (2019). An altitude uncertainty value of ± 10 m is imposed when source publications do not include elevation uncertainty estimates. The past ice thickness site IDs and locations are shown in Fig. 2 and visualized on a site-by-site basis in Fig. S1-S102.

Samples dated using in situ-produced radiocarbon-dated samples were previously not incorporated in the paleo AIS thickness database. Because of the short half-life of $^{14}C$, this method is largely insensitive to inheritance on the deglacial timescales of interest and can therefore help identify cosmogenic nuclide exposure ages unbiased by inheritance. Consequently, in-situ $^{14}C$ dating has resolved inconsistencies in AIS reconstructions for the Weddell Sea drainage sector, where prior cosmogenic nuclide exposure dating suggested hundreds of meters of thinning since the LGM, with neighbouring sites indicating no elevation changes relative to present during the same time period

(Nichols et al., 2019). The inclusion of in situ radiocarbon data from the Shackleton Range, Lassiter Coast, and Schmidt Hills has increased consistency among paleo ice thickness data. Since the LGM, the revised data indicates that the Weddell Sea sector experienced a lowering of the ice surface of ~300 to 600 m above present-day sea level (henceforth referenced simply in meters), with a few sites exceeding 800 m of lowering (Balco et al., 2016; Hein et al., 2016; Bentley et al., 2010, 2017; Johnson et al., 2019; Nichols et al., 2019); this largely reconciles contradictory

reconstructions of the regional post LGM glacial history based on marine and terrestrial records (Hillenbrand et al., 2014).

New exposure data from the Transantarctic Mountains along the Ross Sea embayment tell a more complete, albeit only local post-LGM ice sheet thinning history for the mountain chain. During the LGM, the outlet glaciers presently draining directly into the Ross Sea reached an elevation of 260 to 550 m (Jones et al., 2015; Balco et al., 2019). Of the other outlet glaciers feeding the LGM Ross Ice Shelf/Sheet system, several had an elevation of ~1000 m during the LGM (Spector et al., 2017). Paleo ice thickness data adjacent to the Siple Coast and Ross Island, as originally compiled in the AntICE database showed that the ice elevation at the onset of the post-LGM deglaciation ranged from ~1000 to 2000m. This illustrates the regional variability along the Transantarctic Mountains, where the further

southward, the greater potential LGM elevation changes, with significant variance among the sites likely related to local topographical features of specific valleys (Stone et al., 2003; Todd et al., 2010; Storey et al., 2010).

The Amundsen Sea drainage sector in West Antarctica has limited outcrops suitable for exposure dating; therefore, the region's past ice thickness is poorly constrained. The original database had a total of five data points constraining



the LGM ice elevation in the hinterland of the Amundsen Sea embayment to be between ~500 and 2000 m (Ackert et al., 1999; Johnson et al., 2008). New cosmogenic exposure ages, totalling 25 quality exposure ages, suggest a pre-Holocene ice elevation upwards of at least 900 m (Johnson et al., 2017, 2020). In the original version of the AntICE database, the Antarctic Peninsula lacked any paleo ice thickness data. Three new histories are included in our new iteration, and they all consistently report an ice elevation of ~350 m above present early in the last deglaciation

(Johnson et al., 2009; Bentley et al., 2011; Balco and Schaefer, 2013; Glasser et al., 2014). Finally, a new thinning history from the Sør Rondane Mountains in Dronning Maud Land proposes an ice surface lowering exceeding 1 km during the last deglaciation (Suganuma et al., 2014).

In our study here we also include previously unpublished exposure data constraining AIS thinning since the last glacial

period (SOM Fig. S41, S44, S47, S63). This includes some newer data of high quality (e.g., 1419, 1422, 1425, 1506) that are not yet published in peer-reviewed articles but are included in the ICE-D database due to public access requirements of funding agencies.

## 2.2 Paleo ice sheet marine extent

The stratigraphy of marine sediment cores from the Antarctic continental shelf can preserve some of the complex

history of glacial advance and retreat (Smith et al., 2019). The retreat of the grounding line (GL) can be inferred from the stratigraphic succession from subglacial to GL-proximal glacimarine sediments and that of the calving line can be inferred from the transition of GL-distal glacimarine to seasonal open marine deposits (Smith et al., 2011a; Anderson et al., 2014; Arndt et al., 2017a; Bart et al., 2017; Heroy and Anderson, 2007).

Dating the transition from subglacial to glacimarine facies provides the age of the GL retreat across a core site but usually this approach has to rely on $^{14}$C dating of biogenic material. $^{14}$C dates obtained from calcareous (micro-)fossils provide the most robust age constraints for Antarctic marine sediments. However, there is a paucity of biogenic carbonate in Antarctic shelf sediments in general and in the GL-proximal facies directly overlying the subglacial till in particular. As such, either calcareous fossils (if present) from the open-marine facies or organic matter from the

GL-proximal facies have to be dated (Bart et al., 2017). While the former dates only provide an absolute minimum age for GL retreat from a core site, the latter dating approach is hampered by the fact that the organic matter content in GL-proximal facies is typically very low and that this organic material often comprises large amounts of subglacially-reworked fossil organic carbon. Sometimes resulting in $^{14}$C ages much older than the time of sediment deposition and, thus, the time of GL retreat (Licht et al., 1998; Domack et al., 1999; Pudsey et al., 2006; Heroy and

Anderson, 2007). Over the past two decades, some progress has been made in (i) assessing the reliability of organic matter-based $^{14}$C ages in constraining GL-retreat (Hillenbrand et al., 2010; Smith et al., 2014a), (ii) compound-specific $^{14}$C dating of only the young, fresh fraction of the organic material (Ohkouchi and Eglinton, 2008; Rosenheim et al., 2008; Yokoyama et al., 2016a; Subt et al., 2017), (iii) obtaining reliable $^{14}$C ages from even very small amounts of biogenic carbonate (Klages et al., 2014; Arndt et al., 2017; Erik Arndt et al., 2020), and (iv) utilizing paleo magnetic

methods for dating Antarctic sediment cores (Hillenbrand et al., 2010; Smith et al., 2021).

Retreat of the calving line of an ice shelf is usually reflected in a sediment core from the Antarctic shelf by the transition from a fine-grained terrigenous facies deposited distal from the GL into a biogenic-bearing, often diatom-rich facies deposited under open marine conditions (Livingstone et al., 2012; Yokoyama et al., 2016b; Bart et al.,

2017). However, research on modern sub-ice shelf environments has shown that basin currents can advect biogenic material from open ocean settings far under ice shelves, where they can sustain benthic fauna assemblages and potentially result in deposition of sediments resembling open marine facies (Hemer and Harris, 2003a; Hemer et al., 2007a; Post et al., 2007; Riddle et al., 2007). Measurements of the cosmogenic nuclide $^{10}$Be in marine shelf sediments has shown promise that this ambiguity can be avoided in future studies (Yokoyama *et al.,* 2016). Thus, despite all

the aforementioned improvements, the dating of Antarctic shelf sediments and constraint of the time of GL and calving line retreat still remain a challenge.

The combination of the complex stratigraphy of sediment cores from the Antarctic continental shelf and the lack of reliable age control for key facies renders the interpretation of the proxy record in most cores non-trivial. For this

reason, only those marine sediment records that clearly document a position below grounded ice, under an ice shelf, or in (seasonal) open water at a particular time are added to the AntICE2 database (Fig. 4, Fig. S104-S180).

The paleoEXT database was originally a curated version of the GL retreat ages compiled by Livingstone et al. (2012). Our new iteration has been updated to include the RAISED consortium compilation (Bentley et al., 2014a; Hillenbrand et al., 2014; Anderson et al., 2014; Mackintosh et al., 2014; Ó Cofaigh et al., 2014) and it has also been supplemented by a number of more recent studies (Bart et al., 2018). For each marine sediment core, obvious [14]C age outliers or down-core age reversals, if present, were removed in accordance with the source literature. All the past ice extent ages were recalibrated using a consistent marine reservoir correction of 1144 ± 120 yr (Hall et al., 2010) with CALIB v8.1 (CALIB rev. 8; Stuiver and Reimer, 1993) using the Marine20 calibration curve (Heaton et al., 2020). The full information for the marine cores, including expedition ID, sample depth, etc., is given in the ICE-D marine database (http://marine.antarctica.ice-d.org/).

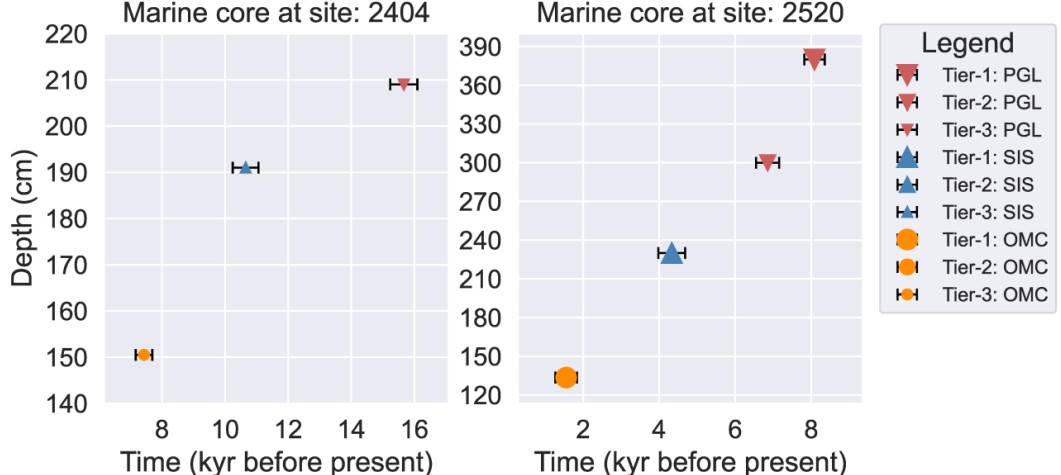

Figure 4: Sample past ice extent (paleoExt – proximal to the grounding line (PGL); sub-ice-shelf (SIS); open marine conditions (OMC)) data from a marine sedimentary core to illustrate the data quality and tier assignment.

Since the original AntICE database, numerous cruises have collected marine sediment cores along transects from near the modern ice shelf front to the continental shelf edge. The biggest addition of data occurred in the Amundsen Sea sector, where LGM grounded ice extent and deglacial GL retreat have been reconstructed across Pine Island-Thwaites Trough (Smith et al., 2011a, 2014b; Hillenbrand et al., 2014; Kirshner et al., 2012), Dotson-Getz Trough (Smith et al., 2011b; Hillenbrand et al., 2013), Abbot-Cosgrove Trough (Klages et al., 2017) and Hobbs Trough (Klages et al., 2014). In Pine Island-Thwaites Trough, the initial GL retreat from the outer continental shelf occurred at 20 ka BP, reaching the mid-shelf by 13.6 ka BP and inner-shelf by 10.6 ka BP (Smith et al., 2014b). In the Ross Sea sector, marine sediment cores indicate an initial retreat from the continental shelf edge prior to the Holocene. The Holocene retreat across large sections of the eastern and western Ross Sea continental shelf was asynchronous (Bart et al., 2018) but occurred during the early to mid Holocene (McKay et al., 2008, 2016; McGlannan et al., 2017; Bart et al., 2018). The calving line of the Ross Ice Shelf retreated throughout the mid to late Holocene, reaching its present extent by ~1.5 ka BP (Yokoyama et al., 2016b). In the Weddell Sea, cores from the outer Filchner Trough suggest the GL advanced and retreated prior to the LGM and readvanced again in the Early Holocene before retreating by 8.7 ka BP (Stolldorf et al., 2012; Arndt et al., 2017a). This additional paleo ice extent data portrays a regionally complex deglacial history (Arndt et al., 2017; Erik Arndt et al., 2020; Hodgson et al., 2018).

## 2.3 Paleo relative sea level

Reconstructions of past sea level are based on a variety of indicators: isolation basins, raised beaches and deltas, marine shells, driftwood, whale, seal and penguin fossils, bedrock exposure dating and lower elevational limits of perched boulders (Verleyen et al., 2005; Shennan et al., 2015; Hodgson et al., 2016; Verleyen et al., 2017). The dated





relative sea level (RSL) proxy data infer an upper bound, lower bound, or a two-way bounded estimate on past sea level given the height of the datum relative to present sea level. Geographically proximal data form a local RSL history which constrains sea-level change through time. Only 0.44% of Antarctica is ice-free land, which limits the regions where past sea-level records can be sampled as well as the availability of certain data types (Hodgson et al., 2016; Verleyen et al., 2017). For the Antarctic domain, the most common RSL data are based on records of raised beaches, isolation basins, molluscs, and penguin remains.

The sea-level proxy data with the highest accuracy are those from isolation basins, which originally formed as marine basins but became subsequently isolated from the ocean through sea-level fall and/or glacial isostatic rebound of the bedrock (N.B.: an isolation basin can later be reconnected to the ocean by subsidence and/or sea-level rise). The sill height that controls drainage from the basin is the RSL elevation proxy. Dating the microfossil remains at the marine-lacustrine/lacustrine-marine transition of a sediment core extracted from an isolation basin determines the age of isolation/reconnection to the ocean. Together, this establishes a precise relative sea-level elevation and age for a given site (Zwartz et al., 1998; Verleyen et al., 2005; Roberts et al., 2011). Past RSL observations of lesser quality that simply constrain a maximum or minimum elevation of past sea level come from $^{14}$C ages on biogenic material buried in raised beaches. Dates on mollusc shells or penguin fossils provide an age for the paleo-beach (Hall and Denton, 1999; Shennan et al., 2015). Similarly, burial ages of raised beaches can be derived from optically stimulated luminescence (OSL) dating of beach cobbles (Simkins et al., 2013). Additional details on the RSL proxy data are discussed in Briggs and Tarasov (2013).

The AntICE2 past RSL sites and their IDs are shown in Fig. 2 and visualized in Fig. 5 and supplementary Fig. S181-S192. Whenever published uncertainties were provided in the source publications, they are incorporated in the database. When they were lacking, a ± 1 m elevation uncertainty is assumed. Moreover, as in (Briggs and Tarasov, 2013), another ± 1 m uncertainty is added to allow for present/paleo tidal variations (Sun et al., 2022) when measured uncertainties are less than 2 m. The radiocarbon ages in the database were recalibrated using the CALIB v8.1 with the IntCal20 (SHCal20), Marine20, or the mixed marine southern hemisphere radiocarbon calibration curve depending on the sample type and marine content (Reimer et al., 2009; Heaton et al., 2020). The source publications use different marine reservoir corrections, depending on the dated material, while our database standardizes the marine reservoir correction to 1144 ± 120 yr (Hall et al., 2010) for simplicity and consistency. By providing the uncorrected $^{14}$C ages, uncertainties, and explicitly including marine reservoir corrections, the relative sea-level dataset can easily be recalibrated. Moreover, this enables the data to be incorporated within an online database (e.g. ICE-D RSL repository), which can be dynamically recalibrated upon request.

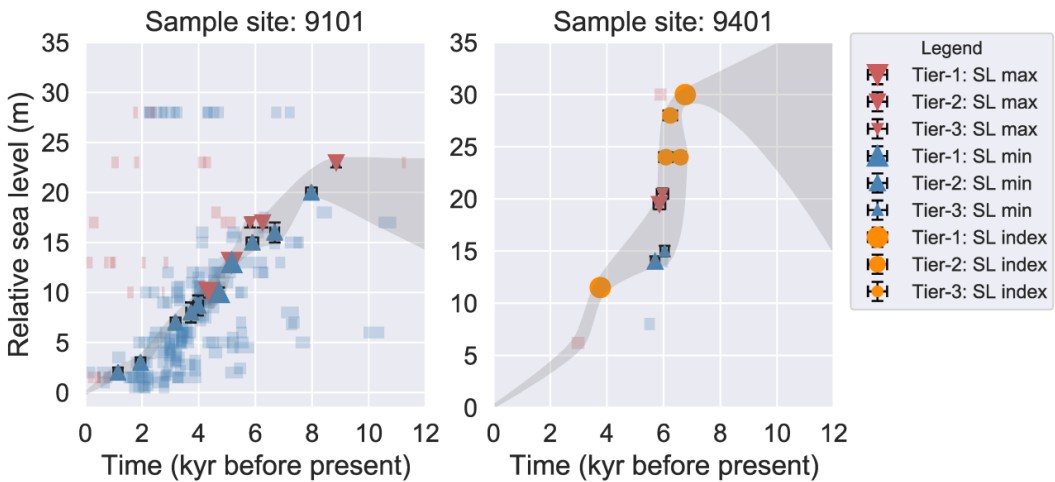



Figure 5: Sample past sea-level (paleoRSL) data to illustrate the data quality and tier assignment. The grey band illustrates the most probable history at the given site. The blue and red transparent bands represent other limiting ages not assigned into a quality tier.

The additional RSL data added in the AntICE2 database have significantly increased the geographic coverage when compared to the original iteration (Simms et al., 2011; Simkins et al., 2013; Hodgson et al., 2016; Verleyen et al., 2017). In Dronning Maud Land, new isolation basin data from Lützow-Holm Bay more robustly constrain past RSL, which is estimated to have fallen by 20 m over the Holocene (Verleyen et al., 2017). These sea-level index points are generally consistent with the previously published limiting dates (Miura et al., 1998). The Lambert-Amery sector around Prydz Bay contains exposed coastal land, where isolation basin contacts, shells and penguin fossils from

raised beaches were dated (Hodgson et al., 2016). This has boosted the reconstructed sea-level history of the region suggesting an early Holocene sea-level rise from -4 to an 8 m highstand at ~8 ka BP, subsequently followed by a gradual fall to PD levels starting at ~8 ka BP (Zwartz et al., 1998; Berg et al., 2010; Hodgson et al., 2016). In the Amundsen Sea sector, there is one date potentially constraining sea-level change based on a [10]Be exposure date from a sample that is suspected to have experienced isostatic emergence from the ocean at 2.2 ka (Johnson et al.,

2008). Alternatively, this exposure date with an elevation of 8 m, could simply reflect ice margin retreat. The Antarctic Peninsula is constrained by six RSL time series. Marguerite Bay provides limiting dates and a few isolation basin ages that indicate a ~20 m sea-level fall from 7 ka BP, reaching PD sea-levels by 1.5 ka BP (Emslie and McDaniel, 2002; Bentley et al., 2005; Simkins et al., 2013). The South Shetland Islands contain some of the largest ice-free sections of land in Antarctica, providing upper/lower bounds on past sea level and, more importantly, isolation basin

index data which imply a sea-level fall from a 16 m highstand at ~8 ka BP (Watcham et al., 2011; Simms et al., 2011). Near and on James Ross Island, isolation basin index data indicate a gradual Holocene sea-level fall, with the earliest constraint indicating sea level was 11 m above present at 6 ka BP (Hjort et al., 1997; Roberts et al., 2011).

## 2.4 Ice core borehole temperatures

The original AntICE database lacked constraints in the interior of the ice sheet. To partly remedy this major issue, we incorporate a new powerful data-type – the temperature profiles of major Antarctic ice core boreholes (Fig. 6). The temperature structure of the ice can be measured by running a temperature logger down the borehole of an ice core (Cuffey et al., 2016). Past changes in temperature, ice velocity, and ice thickness will affect the thermal structure of the ice sheet. Therefore the resulting observations of temperatures through the ice constrain the present and past thermal forcing and ice dynamics.


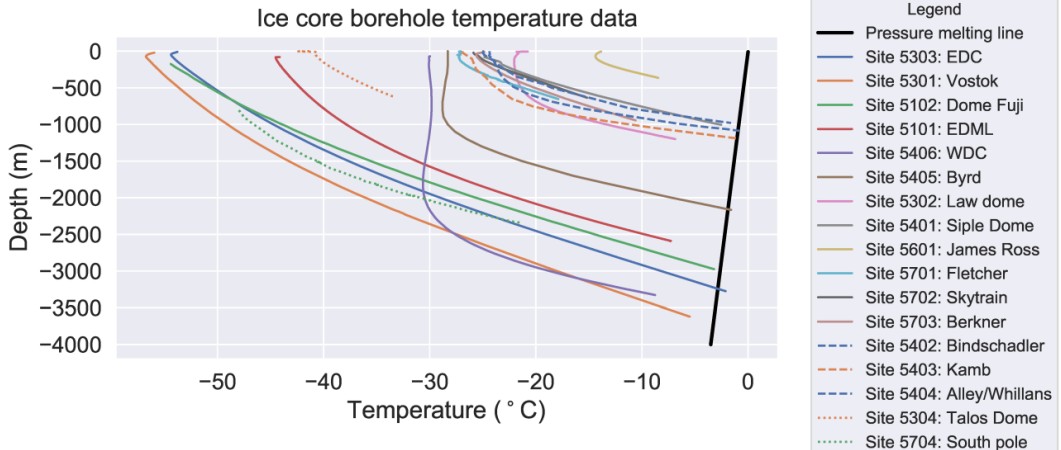



Figure 6: Ice core borehole temperature (boreTemp) data to illustrate the data quality and tier assignment. The dashed lines represent the sites in the Siple coast which are tier-2 data and the dotted lines represent the most limited borehole temperature data which does not cover the majority of the ice column (tier-3 data).


Borehole temperature profiles generally have one of two structures characterized by the depth of the englacial thermal minimum. In the first case, a borehole temperature profile is characterized by minimum temperatures near the ice surface which progressively increase towards the bed (Engelhardt, 2004a, c; Motoyama, 2007; Lukin and Vasiliev, 2014; Weikusat et al., 2017; Mulvaney et al., 2021); this case is typical of low-accumulation sites dominated
by heat diffusion. In the second case, ice temperatures remain cold at depth, and reach a deeper englacial thermal minimum, which is marginally cooler than the surface ice before they warm again towards the base (Gow et al., 1968; Van Ommen et al., 1999; Cuffey et al., 2016); this case is typical of high-accumulation sites dominated by the downward advection of cold surface ice. The ice thickness, geothermal heat flux, horizontal ice advection, and surface accumulation are the main controls on whether or not the base is at the pressure-melting point, with serious
ramifications for basal hydrology and ice dynamics.

The various borehole temperature profiles were measured using different instruments, and some were measured at a time considerably after the ice core had been drilled (Motoyama, 2007; Lukin and Vasiliev, 2014). With the precision of the used temperature logger rarely reported in a source publication, an uncertainty value of ± 0.1 °C is
assumed. The Talos Dome and South Pole borehole temperature profiles do not cover the majority of the entire ice column, minimizing their overall constraint effectiveness capability. Several borehole temperature profiles have been measured along the Siple Coast (Engelhardt, 2004b). Although they all share a high degree of correlation, a total of four Siple Coast boreholes were included in the database to maximize both the spatial distribution and the number of prominent ice sheet features sampled. These temperature profiles are from the Siple Dome borehole as
well as the Bindschadler, Kamb, and Alley/Whillans ice stream boreholes (Engelhardt, 2004b, c).

## 2.5 Present-day uplift vertical land motion

Across Antarctica, a Global Positioning System (GPS) network measures the displacement of the solid Earth. GPS measurements, although relatively scarce, can supplement the even sparser RSL dataset in constraining the isostatic response of the solid Earth to past and present changes in surface AIS load. The vertical deformation rates derived
from GPS measurements represent the integrated signal of several processes operating on various time scales. The two primary contributing factors to vertical land motion are the remaining slow viscous response to past ice and water load changes and the near instantaneous elastic response due to contemporary ice load changes (Martín-Español et al., 2016a; Sasgen et al., 2017).

GPS observations have previously been implemented to evaluate Antarctic GIA and ice sheet models (Argus et al., 2014; Gomez et al., 2018; Whitehouse et al., 2012b; Ivins et al., 2013). The resulting Antarctic GIA estimates are used in conjunction with satellite-derived remote gravimetry or altimetry data to infer contemporary mass balance changes of the AIS (Shepherd et al., 2018).

GPS deformation rates first have to be corrected for the elastic response to contemporary ice mass change before they can be inferred to reflect the background viscous response to past ice mass change (Martín-Español et al., 2016b; Sasgen et al., 2017). For our database, a key criterion for GPS data evaluation is constraint value for Antarctic GIA. We therefore divide the dataset into stations that are not influenced significantly by modern ice mass change and stations with a significant contemporary elastic signal (for details, see the Discussion section below). A total of
67 GPS stations constrain the isostatic adjustment of the land bedrock for the period 2009 to 2014, with a total of 15 GPS stations being assigned into a high quality tier (as discussed in Section 3.1.5).

Alongside the GPS uplift rates, we provide the elastic-response-corrected vertical rates from (Martín-Español et al., 2016a). Both the GPS and elastic-corrected datasets come associated with their own explicit and implicit
uncertainties. In compiling the AntICE2 GPS dataset, we selected sites that are hardly impacted by contemporary mass balance changes (negligible elastic signal). The accuracy of the elastic-corrected high-quality subset of GPS data is dependent on the validity of the inferred contemporary ice load changes and the resulting elastic signal.

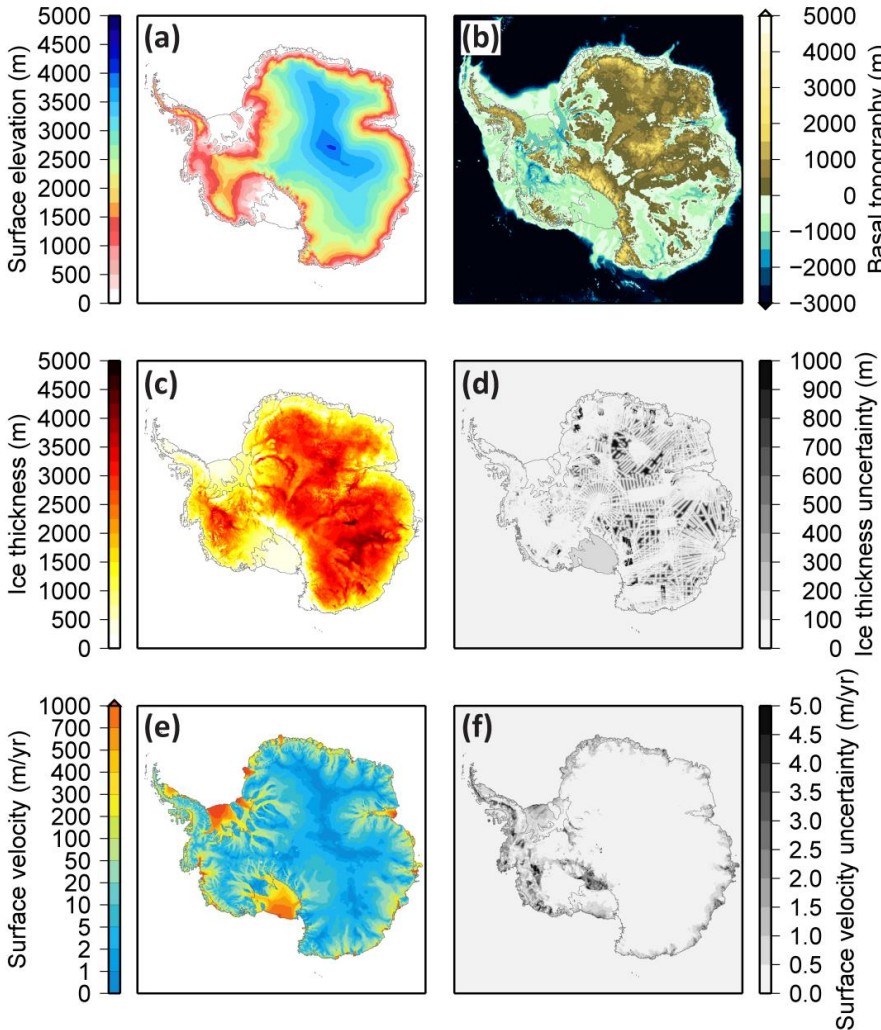

Figure 7: a-d) Present-day Antarctic ice sheet geometry based on the BedMachine version 2 PD data (Morlighem et
al., 2020) and e-f) MEaSUREs ice surface velocity over 2005-2017 and its affiliated uncertainties (Mouginot et al.,
2019).

## 2.6 Present-day geometry and surface ice velocity

The AIS geometry from BedMachine Antarctica version 2 provides the primary PD constraint and initialization
conditions (BCs) (Morlighem et al., 2020). This directly constrains several key PD metrics by comparing the modelled
ice sheet to contemporary observations: ice thickness mean-squared-errors (MSE) for East Antarctica, West
Antarctica, and all ice shelves; squared errors of latitudinal/longitudinal grounding line positions along five key
transects shown in Fig. 2a (Ross, Amundsen, Ronne, Filchner, Amery transect); squared-errors of grounded and total
ice area; squared-errors for the ice shelf area across five sectors (Ross, Amundsen-Bellingshausen, Weddell,
Lambert-Amery, all other remaining sectors combined). These PD observations are powerful spatial constraints but
with limited temporal constraints that only extend back into the late Holocene. The grounding line transects and



sector margins are shown in Fig. 2a. The specific locations of these metrics, particularly the transects, were chosen to investigate areas sensitive to past and present ice sheet changes. The data provided by BedMachine have a horizontal resolution of 500 m by 500 m and includes 2σ uncertainties on ice thickness inferences (Fig. 7a-d). The topographic fields must be upscaled to the appropriate resolution for a given ice sheet model grid; the metrics
discussed above are then calculated at this resolution for consistency. As part of the NASA-funded Making Earth System Data Records for Use in Research Environments (MEaSUREs) program, surface velocities of the AIS have been made available for the period from 2005-2017 (Mouginot et al., 2019) (Fig. 7e-f). The surface velocity dataset is remotely derived from satellite data and provided at a horizontal resolution of 450 m by 450 m, which is similarly upscaled to the ice sheet model grid resolution for data-model comparison and inversion.

## 2.7 Data uncertainty structure

The uncertainties in the database are explicitly stated as 1-σ/2-σ intervals. Some of the observational data in the database exhibit two-way or one-way bounds. It follows that some proxy data and their uncertainties represent just an upper or lower bound constraint (one-way bounds). Two-way Gaussian uncertainties are affiliated with the PD observations (ice sheet geometry and surface velocities), the GPS observations and borehole temperature
measurements. The paleoH and paleoExt data are also represented by two-way symmetric uncertainties around the mean. Some of the paleoExt data constrain exclusively the onset of open marine conditions, rendering them a one-way constraint. There are several one-way constraints in the RSL database as well, such as those that are limiting minimum/maximum RSL inferences (molluscs, penguin remains). The details are specified in the database itself and were previously discussed in greater detail in Briggs and Tarasov (2013). These observations are converted to
nominal two-way non-symmetric constraints by assigning an exceedingly small or large uncertainty bound to the unspecified region of the probability distribution. This adoption of a Gaussian observational error model facilitates ice history scoring. However, the validity of a Gaussian error model for all the types of data in our database awaits future testing.

# 3. Discussion

To implementation of a database for geophysical model calibration has a number of requirements to ensure utility. In large part this boils down to clear specification of the relationship between the data and the real world system under consideration. As such, the database curation process and all data uncertainties needs to be clearly specified. On the modeller side, a careful evaluation of internal and external model limitations is necessary to produce meaningful data-model comparisons. In this discussion, some considerations are explicitly stated when it comes to
the aforementioned challenges.

For much of the last glacial cycle, there are very few to no paleo observations that directly constrain the configuration of the ice sheet (Fig. 8). The paleo data (paleoEXT, paleoH, paleoRSL) have a mean age of 9.5 ka and a non-uniform distribution with a long tail of older ages beyond the LGM. A total of 81% of the paleo data has a Holocene age.
However, some data points integrate ice sheet behaviour over a period and so have constraining power far exceeding their measured age (e.g. PD borehole temperature data - (Ackert, 2003).

The heterogeneity in the spatial distribution of the data is illustrated in Fig. 2. Data of a given type constrain the surrounding region based on the data type and data quality. This is due to the spatial correlation of certain ice sheet
system changes such as margin retreat or GIA. For instance, past ice thickness data might constrain localized ice elevation changes for a particular glacier only. In contrast, past RSL data document changes in the bedrock and geoid elevation, which is a spatially smooth signal. Each data point has a specific spatio-temporal sphere of influence which defines its ability to constrain the model. Figures 2 and 8 illustrates areas in space and time with clear data gaps and densely sampled areas, thus this heterogeneity highlights the importance of never equally weighing all the data
when scoring since it would introduce major biases in a ice sheet model calibration. An inverse-areal weighing of the data can be used to avoid overfitting the model to a particular region with high data density if correlation between data-model residuals is not otherwise accounted for (Tarasov and Goldstein, 2021).



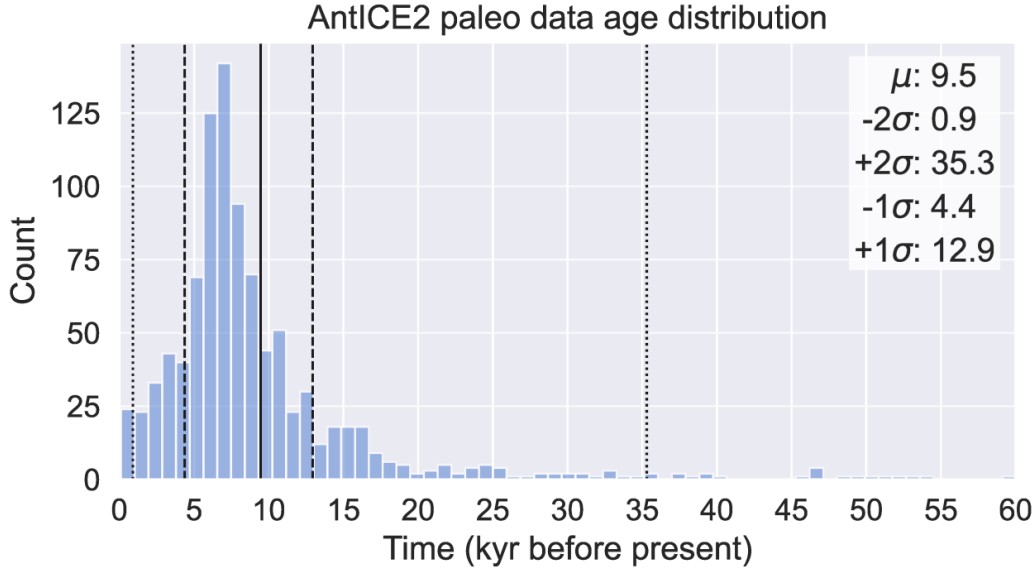

Figure 8: Age distribution of AntICE2 paleo data (paleoExt, paleoH, paleoRSL) where the vertical solid line, dashed lines, and dotted lines represents the mean, +/- one sigma, and +/- two sigma ranges, respectively. The one and two sigma bounds correspond to the nominal 68 and 95% confidence intervals.

Field observations are often collected in Antarctic regions with a complex and highly variable topography, that is inadequately resolved in typical Antarctic-wide ice sheet simulations. Thus, the more a datum embodies broader characteristics of the glacial system, as opposed to reflecting subgrid characteristics, the higher its potential constraint value. For data containing a significant subgrid signal some combination of upscaling of the data and/or downscaling (potentially including sub-grid modelling) of the model results will be required but may not always be physically-justifiable. The RSL change, borehole temperatures, and GPS rates represent spatially and temporally smooth proxies and require no upscaling corrections. The marine paleoExt data capture rather broad non-linear changes in the GL, sub-ice-shelf, or open marine characteristics. However, for GL sites near the continental shelf break, the gridded topography could potentially designate areas at the shelf break as immediately off the continental shelf for which the model would never have a grounding-line. Some terrestrial data, such as nunatak indicators, may also predominantly reflect subgrid high frequency features that won't therefore be resolved by the model. Given fundamental model limitations, such as grid resolution, for most, if not all data, the physical signal represented by the data (i.e. after accounting for observational uncertainties) will only be incompletely resolvable by the model even after appropriate upscaling and/or downscaling. The resultant fundamentally irreducible discrepancy between the model results and geophysical system will then need to be accounted for in the error model describing the relationship between the model and physical system (see Section 3.1.8).

The uncertainties associated with the indicative meaning of various proxy data must be considered when estimating observational uncertainties. This is particularly relevant for the paleoExt data because ocean currents can advect particles from open water beneath an ice shelf, so that the resulting deposits resemble open marine facies. This can be up to 6 km and 100 km from the calving front for small and large ice shelves, respectively (Riddle et al., 2007; Post et al., 2014; Hemer and Harris, 2003b; Hemer et al., 2007b). Similarly, facies characterizing sediments deposited proximal to the GL (PGL) can form up to 10 km seaward from the GL at the time of formation (Smith et al., 2019). When it comes to paleoRSL data, considerations should be made for the fact that storm surges can impact the in-situ deposition of certain proxy data. This has previously been handled by applying a storm beach adjustment factor of up to 1 m for proxy data from, but not limited to, mollusc fragments and penguin bones (Briggs and Tarasov,




2013). This information is not integrated within the AntICE2 database, and we defer the incorporation of such uncertainties within a data-model scoring implementation.

## 3.1 Data curation and tiered data quality assessment

To facilitate data-model comparison, the paleo ice extent, ice thickness, relative sea level, borehole temperatures and PD uplift rates are all curated, and individual data points are categorized into quality tiers (Table 1). Tier-1 is the
highest quality tier while tier-3 is the lowest accepted tier. Tier-1 data has the most constraining power on the ice sheet and GIA history. For example, in the case of cosmogenic exposure dating, key exposure ages capturing both the LGM ice thickness and timing of deglaciation prove to be most valuable since this constrains the most prominent deglacial changes. Tier-2 data typically represent data with less constraining power that primarily supplement Tier-1 data. Returning to the previous exposure data example, tier-1 data represents a minimal set of crucial tie points
for the LGM ice thickness and timing of deglaciation, while tier-2 data provide finer detail for the deglacial ice sheet thinning history, with minimal correlation to other data. Finally, tier-3 data include lower quality observations that exhibit a high degree of correlation with tier-1/2 data and for which data uncertainty specifications are less confident. This tier assignment depends on data type, availability, and data density. This particularly becomes an issue at sites with limited observations. When mentioning lower quality data, we refer to data with larger
measurement/analytical uncertainties or limitations due to a proxy's indicative strength (i.e., whether its interpretation is ambiguous or not). Data that are not assigned to a quality tier typically represent redundant data, data with very large uncertainties, or data which do not accurately represent the local environmental history, or where original publications note potential analytical problems. In the case of cosmogenic nuclide data, exposure ages that clearly suffer from inheritance are not be assigned to a quality tier. Moreover, some data is excluded from
tier assignment based on physical impossibility – e.g. exposure age data require that younger ages must be at a lower elevation than older ages for a given site. In this section, we describe the tier assignment process which involves evaluating the datasets with respect to strict criteria that assign the dataset into their respective tiers. Further refinements are conducted based on expert assessment and outlier identification. This is performed on a data type basis and the aim is to minimize subjectivity in quality assessment and data selection. There are a few criteria which
are consistent across data types: prioritizing data with a high signal to noise ratio beyond a chosen data type threshold; valuing data in data-sparse regions; outlier identification and exclusion when substantiated by broadly consistent, dense data clusters (cluster density assessments are data type dependent); and superfluous data exclusion.

Table 1: Summary of Antarctic ICe sheet Evolution database version 2 (AntICE2) and quality tier subsets.

| Datatype | All data | Tier-1 data | Tier-2 data | Tier-3 data |
|---|---|---|---|---|
| Past ice extent (paleoExt) | 249 | 63 | 15 | 30 |
| Past ice thickness (paleoH) | 2710 | 108 | 348 | 270 |
| Past relative sea level (paleoRSL) | 425 | 23 | 48 | 52 |
| Ice core borehole temperature (boreTemp) | 36740 | 12x3 | 3x3 | 2x3 |
| Present-day GPS uplift rates (rdotGPS) | 67 | 6 | 9 | NA |
| AntICE2 | 40191 | 236 | 429 | 358 |

## 3.1.1 PaleoH data curation

The past ice thickness dataset requires additional considerations when assessing data tiers. Some paleoH sites have few exposure ages that constrain the elevation history. In these instances, we rely on the discretion of the original
study to assess the quality of the data point which is available through ICE-D. At a given site, an assessment is conducted that identifies the highest quality exposure ages (e.g., $^{10}$Be, $^{26}$Al, $^{3}$He, $^{14}$C) that bracket the elevation history and sorts the data into tiers (Fig. S1-S102). A high data density cluster of young exposure ages that form the most probable elevation history is identified (Fig. 3). This assessment considers the occurrence of inheritance and post deglacial shielding. By evaluating paired isotope exposure ages and applying first principles (along a sample
transect older ages should always be obtained from samples collected at higher altitudes) many data points can be



excluded and a most probable elevation history based on data density can be identified. Tier-1 data are the data constraining the magnitude, timing, and rate of elevation change over the deglaciation. Tier-2 data further constrain the specific structure of the elevation history. Tier-3 data include the remaining pertinent data, which fully populate much of the most probable elevation history. The primary reason for tier-3 data to be relegated to their own tier,

rather than to include them in the tier-2 data, is the limited constraining power that they introduce to past ice thickness changes given how they correlate significantly with tier-1 and tier-2 data, which renders them nearly superfluous in many instances. When evaluating a cluster of neighbouring sites, a certain degree of consistency should be expected, if the exposure data are truly representative of a broader region rather than extremely local ice elevation changes. Therefore, an additional iteration on the tiers is performed, based on upstream/downstream

consistency to neighbouring paleoH sites, to identify potential outliers. The source literature of Antarctic exposure ages does not always report the sample position relative to the mean flow of the surrounding ice. This proves to be an issue when comparing ice thinning histories reconstructed by continental-scale ice sheet models with histories based on paleoH data because the exposure ages can be heavily biased depending on the nunatak flank, where the samples were collected (Mas E Braga et al., 2021). This information is not broadly accessible in the source literature,

causing a limitation which propagates into the AntICE2 database and it must be considered within the error model when scoring a reconstruction against paleoH data.

### 3.1.2 PaleoExt data curation

The past ice extent dataset is also divided into tiers based on specific data-type criteria. Firstly, the interpretation of the facies is extracted from the source literature and assigned a class (Fig. 4 and Fig. S103-S180) from: proximal to

the GL (PGL), sub-ice-shelf (SIS), or open marine conditions (OMC). Each core is then sorted according to the [14]C dating method, i.e. whether the [14]C age is obtained from biogenic carbonate or organic matter, with the latter dates typically considered to be less reliable (see section 2.2). Down-core [14]C ages obtained from organic matter in sediment cores from the Antarctic continental shelf cores are often corrected by subtracting the core top age rather than the marine reservoir effect only (e.g., Domack et al., 1999; Pudsey et al., 2006). This approach assumes that the

degree of "contamination" of young organic carbon with reworked, fossil organic matter has remained constant throughout the record, which often, however, is not the case (e.g., Heroy & Anderson, 2007). For these reasons, [14]C ages on calcareous microfossils, if present, are typically favoured over organic matter [14]C ages, and the former are typically assigned to a high-quality tier. Additional criteria toward sorting the paleoExt data into tiers are based on the overall quality of the marine sedimentary record and facies interpretation, specifically, whether the stratigraphy

of the core is affected by reworking (e.g., due to iceberg turbation). Moreover, if in a given core multiple dates were obtained from different facies that indicate the same environmental conditions, the maximum and minimum dates bracketing the age cluster are assigned to a high-quality tier, whereas the remaining dates are excluded to avoid redundancy. These criteria are enforced when assigning tiers to the marine paleoExt data and deciding, whether to exclude ages from direct data-model scoring.

### 3.1.3 PaleoRSL data curation

Compared to other data types, there are limited past RSL observations. For this reason, the quality assessment of the paleoRSL data is performed on a site-by-site basis. For a given site, we define the most probable RSL history as constrained by sea-level index points and min/max bounds (Fig. 5 and Fig. S181-S192). This approach inherently identifies potential outlier data for exclusion. Tier-1 data for a site comprise the highest quality proxy data that constrain the highstand and the form of the deglacial sea-level fall. Data that constrains the RSL history with minimal

redundancy and supplements tier-1 data are assigned tier-2 status. The classification of tier-2 data is based on data density along the most probable theoretical RSL history (Fig. 5). Tier-3 RSL data further populate the most likely RSL history already defined by the tier-1/2 data and, they consist of lower quality constraints that correlate to tier-1/2 data without being completely superfluous.

### 3.1.4 Borehole temperature data curation

The ice-core borehole temperature profiles consist of a significant amount of data along a single profile, much of which is highly correlated with depth. Therefore, a subset of the profile data is chosen and assigned tier-1 quality for data-model scoring. The tier-1 data consist of the coolest near-surface ice temperature, the nearest basal ice temperature, and the ice-column midpoint englacial temperature. These data alone can effectively constrain the structure of the simulated temperature profile given the smoothness of the signal. South Pole and Talos Dome

borehole temperature profiles were the only profiles that do not cover the majority of the ice column (Fig. 6).



Therefore, they provide less constraining power on a model and are assigned a tier-3 status. The Siple Coast borehole profiles from Siple Dome, Bindschadler, Kamb, and Alley/Whillans ice streams are relatively proximal and correlate with each other, so they are assigned a tier-2 status apart from the Siple Dome profile which remains the regional
tier-1 representative. The remaining ice core borehole temperature profiles (solid-coloured lines in Fig. 6) are all part of the tier-1 subset because of the quality and location of the data (Fig. 2d).

### 3.1.5 Uplift rate data curation

PD uplift rates inferred from GPS observations constrain several integrated processes. Prior to sorting the GPS uplift rates into tiers, the GPS data must be evaluated to identify a subset which is most suitable to constrain the GIA signal
and hence past ice load changes. First and foremost, contemporary ice sheet change triggers an elastic response that contributes to the PD observed uplift rates and hence masks the signal associated with past ice sheet change. GPS data from sites where significant PD elastic contributions were inferred (Fig. S193), are considered as low quality constraints on the contemporary GIA signal and hence past ice load changes. Several criteria, such as a low elastic correction (<0.55 mm/yr) and a small uplift rate standard deviation (<1 mm/yr) or high signal to noise ratio (>1.45),
determine which GPS data are considered for data-model evaluation and tier classification. Additionally, GPS sites that are in the vicinity of the coast (<250 km) or in areas, where significant mass loss has evidently occurred over the last millennium, are excluded from being classified into a tier. The GPS data that pass these criteria are assigned a preliminary tier-2 status. A final criterion considers a common model limitation, which pertains to the spherically symmetric viscosity profile of many GIA models and excludes the presence of lateral viscosity structure. This criterion
can be disregarded, if dealing with a 3D Earth viscosity model. In several regions of West Antarctica the continental crusts is underlain by a mantle with anomalously low viscosities in the top 250 kilometers (Whitehouse et al., 2019). GPS sites near anomalous viscosity features are more capable than others of biasing the model calibration given the structural uncertainty associated with the GIA model. Therefore, certain sites are identified along parts of the Antarctic Peninsula, and in the Amundsen-Bellingshausen Sea sector and Ross Sea sector, where the inferred
viscosity at depth is below $10^{20}$ Pa·s (Whitehouse et al., 2019). These criteria filter the GPS uplift rate data based on their quality and ability to constrain the GIA signal and past ice load changes. One persisting issue is the robustness of the elastic corrections, which are likely to have underrepresented uncertainties (Martín-Español et al., 2016a). Uncertainties in the elastic corrections can be increased by the root sum square with the elastic correction to boost confidence that the elastic-corrected uplift rates accurately constrain the viscous GIA response. Of the tier-2 GPS
sites, those which are not located in regions with anomalously low mantle viscosity, are promoted to tier-1 quality status (6 sites). The exact thresholds for the various criteria are based on the need to identify a higher-quality data subset, while simultaneously accounting for unquantified uncertainties associated with the elastic corrections. The criteria-defined higher-quality subset size is chosen to represent the top third of tiered GPS data (tier-1/2 data), offering sites that are especially sparsely distributed. Refinements to the criteria thresholds will be required as the
size of the GPS network evolves and as more robust approaches to interpreting GPS time series are developed.

### 3.1.6 Present-day AIS geometry data curation

The PD AIS geometry and surface velocities are crucial constraints that provide nearly complete spatial coverage rendering it tier-1 data. Regions with large uncertainties in the PD AIS bed geometry and surface velocities could be classified as tier-2 data points, however given these regions typically have no other data constraints they remain top
tier data. For ice sheet modellers scoring their simulations against them, it is important to account for the uncertainties in these inferences when calculating a root mean square error score. It has been shown that spectral noise models, which introduce spatially correlated noise, can be used to produce an ensemble of boundary conditions that are self-consistent with the underlying field and uncertainty estimates (Sun et al., 2014; Gasson et al., 2015). This provides a method allowing for a proper quantification of uncertainties affiliated with these boundary
conditions.

### 3.1.7 Data standards and expert assessment

Fundamental and recurring issues, which exacerbate the challenges of evaluating data quality, remain across many studies and data types. They relate to the data availability. For example, some studies make the entirety of uncorrected, corrected and calibrated $^{14}$C ages available (Heroy and Anderson, 2007; Bentley et al., 2014b), while
others provide only those with robust interpretations (Domack et al., 2005). This makes it challenging to assess the entirety of a broad dataset by the same standards since some data relies on the implicit assessments internally made

by their respective study. Ideally, all associated data should be made available, and the data should be categorized into quality tiers based on explicitly specified criteria. The expert quality control by the principal investigators who collected the samples and analysed and evaluated the data is exceedingly valuable and should be included with the

data. This enables a broader consensus on quality control as various experts converge towards specific quality criteria. Moreover, when new proxy data of various quality are introduced in the future, including potential novel constraints, it will be possible to re-assess the categorical quality tiers, if their criteria are clearly specified. The data made available should contain all the necessary information to recalibrate the data with clearly specified uncertainties. This will facilitate future data calibration, standardization, and quality assessment once integrated

properly within an online repository, such as the ICE-D repository (Balco, 2020).

### 3.1.8 Data-system and system-model uncertainties

The data presented in the AntICE2 database includes data-system uncertainties, typically referred to as observational uncertainty which consists of measurement uncertainty and indicative meaning uncertainty. The former represents uncertainties affiliated with inherent instrumental uncertainties when taking measurements,

such as the elevation at which a cosmogenic sample is collected. The latter being uncertainties relating to the interpretation of data and how it represents a proxy observation for physical characteristic of the system, for instance a fossil mollusc fragment and how it relates to past sea level. In the AntICE2 database, we include the indicative meaning uncertainty from the source literature when available and otherwise do not attempt to specify it. On the other hand, we do specify a baseline measurement uncertainty when absent in the source

publication or when clearly understated.

As the observational uncertainty specifies the data-system relationship, meaningful data-model comparison also requires specification of the relationship between the model and the physical system. However, appropriate specification of the structural error model is a major challenge. The source of this challenge is that we cannot have

complete knowledge of the current and especially of the past state of the earth system or any significant sub-component thereof. As such, we cannot easily identify and quantify model deficiencies with respect to the system of interest. There are many approaches for dealing with these challenges and we point the reader to Tarasov and Goldstein (2021) for a broader discussion.

## 3.2 Potential future and rejected data types

Radiostratigraphy of the Greenland ice sheet has been used to infer the age structure of the ice sheet (MacGregor et al., 2015). Proof of concept age tracking simulations of a 3D slice through Greenland summit have demonstrated the potential constraint value of such data (Born, 2017). The age structure of the ice is inferred from internal reflectors (reflective isochrones) visible in the radiostratigraphic profiles, which are dated at major ice core sites (Cavitte et al., 2020). The AntICE2 database does not include any direct age structure constraints for the Antarctica

Ice Sheet. It would be extremely valuable to have an age structure for the entire AIS because this would provide constraints for many regions that are lacking any paleo constraints. However, the presently-available radiostratigraphy coverage for the AIS is spatially limited and as such, no such AIS-wide reconstruction currently exists to date. There are some regional age reconstructions for well-studied regions and transects (Ashmore et al., 2020; Winter et al., 2019; Delf et al., 2020; Cavitte et al., 2020). A compilation of age reconstructions has been started

under the AntArchitecture initiative within SCAR (https://www.scar.org/science/antarchitecture/antarch-news/). Sutter et al. (2021) demonstrate a data-model comparison of various age isochrones in Antarctica and illustrate the utility of this new data type. As accurate ice age tracing modules become available for ice sheet models, radiostratigraphic age constraints will provide a powerful new constraint on past AIS evolution.

Ice cores have previously been used to infer past AIS elevation changes relative to PD. Originally this was done by analysing the gas content trapped in the ice (Lorius et al., 1984; Delmotte et al., 1999), relating the total gas content to the ambient atmospheric pressure at bubble close-off. This traditional method produces a high noise to signal ratio, especially because other processes affect the volume of open pore space in the ice, such as insolation (Raynaud et al., 2007) and climate (Eicher et al., 2016). We therefore do not include air content observations in the AntICE2

database. An alternative method to determine past elevation changes at ice core sites is through model inferences, where model simulations are locally constrained by ice core data (Barbante et al., 2006; Neumann et al., 2008; Steig et al., 2001; Waddington et al., 2005; Parrenin et al., 2007). The issue with using model inferences to constrain a



model is that they integrate all the assumptions involved in making those inferences, and these are often not explicitly specified. Moreover, the uncertainties in the ice core site elevation model inferences are often inadequately explored and hence underestimated, and would benefit from a greater exploration of the range of uncertainties (Steig et al., 2001). If included in a calibration, this would propagate ill-defined uncertainties and could invalidate the calibration. Therefore, we opt to exclude this dataset since it is too far removed from direct field observations and comes associated with significant and ill-defined model uncertainties.

As previously mentioned when discussing the GPS data, geodetic observations have been used to constrain the GIA response signal associated with past ice sheet changes (Martín-Español et al., 2016b; Sasgen et al., 2017). The justification for excluding the inversion-based Antarctic GIA reconstructions as a constraint lies in the assumptions behind the elastic corrections associated with contemporary mass loss. Like the ice core-inferred elevation changes, the elastic corrections come associated with ill-defined model uncertainties which could invalidate the calibration, if implemented without expanded uncertainties. Moreover, if the intention is to calibrate an ice sheet and GIA model to infer contemporary mass balance by correcting geodetic data, it would be a circular reasoning to apply a constraint that makes a priori assumptions about the form of the GIA signal.

Several ice cores have been drilled to the bed across the Siple Coast and the basal till that has been retrieved containing contains organic material that yields $^{14}$C ages significantly younger than 40 ka BP but older than 20 ka BP (Kingslake et al., 2018). This poses the question of whether the GL retreated landward of the core sites during the most recent deglaciation. The presence of organic matter with a last glacial period age at the base of the modern AIS is hard to reconcile because all major continental ice sheets, including the AIS, are believed to have reached their maximum extent and size during this time. Subglacial sediments contain mixtures of eroded and reworked detritus initially deposited at different times. Therefore, a $^{14}$C date obtained from the organic matter of such subglacial sediments typically provides an integrated age, derived from the mixing of relatively young with old and even $^{14}$C-dead material, which increases the uncertainty of how to interpret such an age. Kingslake *et al.*, (2018) opted to use the $^{14}$C dates as evidence of an early Holocene GL retreat upstream of its PD position, thereby arguing that the $^{14}$C dates do not represent true ages for sediment deposition ages. Given these uncertainties, and notwithstanding further studies (Neuhaus et al., 2021) on the $^{14}$C dates from the till samples along the Siple Coast, they have not been included in the AntICE2 database as paleoExt GL retreat constraints because they do not yet represent a firm and reliable age constraint on GL position.

The main outstanding issue with the AntICE2 database are the temporal and spatial data gaps. As shown in Fig. 1, 2, and 8, only a small number of dates extend beyond 20 ka, and the spatial distribution of the data leaves many regions, particularly in East Antarctica, completely devoid of observational constraints. The ramification of the data "deserts" is that calibrated models will likely have large uncertainties in regions with limited observational constraints. A few new data types, discussed above, could ameliorate the situation, with the most promising being the wide-scale age structure of the AIS, inferred from airborne radar mapping of internal layering connected to sites of dated ice cores. This data type could constrain changes in the AIS far beyond 30 ka and even cover regions with little or no data due to a lack of rock outcrops and boreholes.

Future work should lie focus on using calibrated model results to establish an "Antarctic treasure map" similar to those produced for ice cores by Bradley et al. (2012) that identifies high priority targets for the collection of observational data (Tarasov and Goldstein, 2021). Such a map would highlight the constraining power of various hypothetical observational constraints, for example, those taken from unsampled nunataks or paleo-grounding zone wedges preserved on the continental shelf. Finally, this future progress crucially depends on the growth of well-maintained online data repositories (e.g. ICE-D, Ghub), the careful curation of data, the standardization of the curation criteria, and the proper methodological approaches toward data-model comparison.

# 4. Data availability

The supplementary section contains plots of the entire tiered AntICE2 database. Summary plots provide concise representations of various data types when possible. The AntICE2 database can be downloaded as Excel tables





## 5. Summary

In this study we provide the second major iteration of the Antarctic ICe sheet Evolution observational constraint (AntICE2) database. The AntICE2 database is a curated observational constraint database intended for the calibration of models of the Antarctic ice sheet and Antarctic glacial isostatic adjustment over the last glacial cycle. It can also be used to constrain a paleo model spin up of the AIS to initialize PD simulations. The AntICE2 database includes a large variety of observational constraints necessary for model calibration. The data types included are as follows: PD geometry and surface velocity, PD uplift rates, borehole temperature profiles, past ice extent, past surface elevation, and past relative sea level. All the $^{14}$C ages in the database are recalibrated and share a consistent reservoir age correction, wherever appropriate. The AntICE2 database represents a curated dataset with specified quality tiers. This was achieved by establishing and applying criteria for the different data types. Future efforts should be geared toward refining the criteria for the quality tier assignment since a community consensus would benefit data-model integration. An ongoing effort involves automating the selection and curation process from a raw database (ICE-D) to a recalibrated curated subset (i.e. AntICE2). This would render the AntICE2 database more manageable and updatable as more data is collected in the future. To contribute to the AntICE2 database, one can contact the corresponding author with data/publications, contribute data to the ICE-D databases, or offer data type criteria modification to help revise the data curation process. The AntICE2 database represents the most comprehensive observational constraint datasets of high-quality data relating to the past evolution of the Antarctic Ice Sheet. The dataset facilitates data integration with Antarctic ice sheet and GIA simulations. This dataset compilation also facilitates data-model scoring by processing and curating large raw and disparate datasets from online repositories (e.g., ICE-D) and source publications. Finally, a call to the community is made to make raw data with complete and clearly specified uncertainties publicly available and to make efforts towards establishing data quality criteria in order to facilitate data curation and hence produce meaningful data-model comparisons.

## Author Contribution

B.S.L. and L.T. led and designed the study. B.S.L. compiled and recalibrated the datasets. B.S.L. wrote the manuscript and generated all the figures. G.B. and P.S. contributed past ice thickness and past ice extent data via ICE-D. C-D.H. contributed some past ice extent datasets and assisted in its interpretation. C.B., C.R., M.L-L, and R.M. contributed some ice core borehole datasets. P.L.W. and M.J.B contributed some relative sea-level datasets. J.B. provided the GPS datasets. All authors participated in establishing the data-quality criteria to curate the database and provided feedback on the manuscript.

## Competing interests

The authors declare that they have no conflict of interest.

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
