# Peer review of "Antarctic ice sheet paleo-constraint database"

_Earth System Science Data, 2022_

## Author Comment (AC2)

We would like to thank Torsten Albrecht and Steven Phipps for their referee comments, suggestions, and feedback. This response aims to address any comments raised by the referees. Our responses are embedded below and are shown in orange.

**Response to referee comments #1 (Torsten Albrecht)**

**General comments:**

Lecavalier et al. present a compilation of 7 different observational data types for the Antarctic Ice Sheet covering a period from present day, Holocene and the last deglaciation with a sparse coverage of the last glacial cycle. The database is made available as an Excel spreadsheet (via the open data repository Ghub and/or as Supplement) and can be used to constrain Antarctic Ice Sheet reconstructions, or in general to calibrate paleo model spin-ups, as basis for stability analysis or future sea level projections. The database also encompasses two present-day data sets for the ice sheet and bed topography (BedMachine Antarctica v.2, Morlighem et al., 2020) and surface velocities (MeaSUREs, Mouginot et al., 2019), both available at NSIDC (free NASA Earthdata Login account is required to access these data), and several related key metrics are named.

The AntICE2 database is an extension of the AntICEdat, published 10 years ago by Briggs and Tarasov (2013), and comes with specified observational uncertainties. Briggs et al., (2014) proposed a methodology by which the database can be applied to evaluate (score) an ensemble of modeled Antarctic Ice Sheet reconstructions by using observational error models and data-weighting to address the heterogeneous distribution of the data in space and time. To my knowledge, a few ice sheet model ensemble studies used a subset of the first iteration database (in combination with GPS and RAISED data, e.g. Pollard et al., 2016, Albrecht et al., 2020b) to evaluate model-data misfit. But I very much support the need for a general integration of such a constraint database and the systematic exploration of structural model uncertainties in a proper uncertainty assessment (Tarasov and Goldstein, 2021).

The updated dataset has been improvement in (at least) three ways: Two new data types (GPS and borehole temperature profiles) have been integrated, newly available data have been merged (more than 1000 direct observational constraints), and the whole dataset has been consistently re-calibrated and curated (quality- assessed). Some of the data types are based on various source publications, for instance the relative sea level indicators (paleoRSL), with valuable updates in locations like Dronning Maud Land and the Amundsen Sea sector. Other data types build on other open databases, for instance exposure histories (paleoH), which are based on the informal and inclusive ICE-D database, which still suffers from inconsistencies. Grounding line retreat proxies (paleoExt) are based on the ICE-D marine database and now include the RAISED consortium compilation (Bentley et al., 2014a).

The data description paper explains the selection and quality criteria adequately. It is really helpful for modelers, in particular as is hard to find, sort and evaluate the quality and consistency of the data quality, when not trained in geology. The corresponding article describes the evaluation processes in a well-structured form. The figures are informative. I would have some minor practical recommendations on the structure and format of the dataset (see below).

AntICEx may become an ongoing project open for new data to come. The article addresses potential future data types and reasons for rejecting data types. It also suggests standards of experts quality control on new proxy data, which would simplify the process of updating datasets. I highly support the publication of the AntICE2 database, this is extremely valuable work!

We would like to thank Torsten Albrecht for his referee comments.

**specific comments:**

The paleo data is categorized by site (four digit identifier for data type, basin sector, site within each sector) and include location data (latitude, longitude), age data and an indication of the relationship between the proxy observation and the constrained characteristic.

From a modelers perspective there are quite some steps to take from the pure dataset to the evaluation of model simulations or even model calibration. And I wonder what could simplify this processing already on the data level. For instance, the most relevant data columns (e.g. SITID, SSID, DBCD, LON, LAT, VAL, VALU, ACAL, ACALL1U, ACALU1U, CTY, TIERS) could be re-ordered in a consistent way for each data type.

      The format of an Excel Sheet (with size of < 1 MB) is easy to access for a wider community, but there might by
other formats (e.g. the human-readable JSON), which are easier to access with data processing tools in python etc. There are also quantitative tools like the Automated Timing Accordance Tool (ATAT version 1.1, Ely et al., 2018), which use gridded geochronological data to evaluate model-data misfit (e.g. with a GIS software package). There are obvious shortcomings in the accuracy of this approach, but such an intermediate step may help to bridge first obstacles for potential data users. The focus of this paper is on the data selection, but such practical applications
may be worth to mention.

      The data was assembled and processed in python, the spreadsheet format was used to maximize accessibility. The ordering of the columns is based on differences arising between datatypes but efforts were taken to maximize consistency. The spreadsheet can easily be simply saved as a csv file which can be read into python via a pandas dataframe using a simple command (e.g. df = pd.read_csv( AntICEdatv2.PaleoExt.v8.csv, delimiter=',')). The data
can then reordered and filtered (e.g. paleoExt = df['SITID', 'LON', 'LAT', 'CLASS', 'ACAL', 'ACALL1U', 'ACALU1U', 'TIERS']; paleoExtTier1 = paleoExt[ paleoExt['TIERS'].isin([1]) ] ). Pandas is one of the most popular python libraries in data science and makes handling the AntICE2 quite straight forward. A pandas dataframe can be converted to json using:
      # convert the dataframe to a JSON string
json_str = df.to_json()
      # write the JSON string to a file
      with open('data.json', 'w') as f:
          f.write(json_str)

      The authors are unfamiliar with ATAT but AntICE2 can be hooked into it using pandas dataframes to facilitates
data-model comparison. A reference to ATAT has been added to the revised manuscript: "There are many sophisticated approaches to perform a meaningful data-model comparison (Tarasov and Goldstein, 2021) and there are tools that can assist those wanting an initial, albeit limited data-model comparison implementation (e.g. Ely et al., 2019)."

      The databases also encompasses two present-day data sets for the ice sheet and bed topography (BedMachine
Antarctica v.2, Morlighem et al., 2020) and surface velocities (MeaSUREs, Mouginot et al., 2019). Both datasets are available at NSIDC (free NASA Earthdata Login account is required to access these data), a link should be added to the article, or at least to the citation. From the article it is not immediately clear, that the two datasets are just selected to be part of AntICE2 with assigned tiers and several key metrics named (maybe add the reference to Briggs et al., 2014 here), but are not actually provided for download (as part of this data publication). The
BedMachine data are already available as Version 3 (https://nsidc.org/data/nsidc-0756/versions/3, https://nsidc.org/data/nsidc-0484/versions/2).

      A link is provided in the text to the Bedmachine and MeaSUREs datasets on nsidc.

      The article describes three different quality levels (tiers). What is lacking is some kind of recommendation, which minimum classification to use in model evaluation. Should one avoid tier 0 (and -1) in any case? For instance, can we trust the 6 tier-2 exposure ages at site 1701, spanning 300m of ice thickness change within 2kyr time? Or should only the highest quality (tier 1) be considered?

Additions to the text have been made in Section 3.1 to provide guidance on which tiers to use for data-model comparison. For example, tier = -1 should be excluded in everything, while tier = 0 should not be used for scoring but can be used for visualization. Tier 1, 2, and 3 can be used in data-model scoring, however, by including tier 2

and 3, you are increasing the correlation between datapoints which complicates the likelihood function and spatio-temporal bias.

**technical corrections:**

l.71: "… the PD ice sheet geometry (surface elevation, ice thickness, basal topography) and surface ice velocities are provided." → Better indicate, that those datasets are available elsewhere and have been selected in the quality assessment.

Revised accordingly.

Fig. 3B: How is this uncertainty range (grey shading) constructed?

It is the 2-sigma bound at the sites based on expert assessment and first principles. It is used here for illustrative purposes to convey how the tier classification coincide with an expert assessment to help define the quality criteria. A mention of this has been added to the figure caption.

l.150: Provide a URL to ICE-D for consistency.

Revised accordingly.

l.172: When referring to certain data locations in the text (e.g. the correctd paleoH in the Weddell Sea sector), please provide the site ID?

Revised the text where this occurred. In this instance this referred to sites: 1701, 1713, 1715

l.223: Please reorder to formulate a sentence?

Revised to: "This can result in $^{14}$C ages much older than the time of sediment deposition and, thus, the time of GL retreat."

l.280: It could help readers without a background in GIA modeling to mention that RSL comprises both the bed rock uplift but also the near-field and far-field sea level changes in response to global ice sheet changes, and that after the deglacial RSL rise, data are only available in the following period of delayed bedrock adjustment.

Anyone working with or modelling RSL already understand this. GIA can potentially cause RSL fall or rise in different locations around Antarctica during the Holocene. Moreover, GIA is not the only cause of RSL change and summarizing the various causes of RSL change is not necessary in the context of this, already plenty long manuscript.

l.304: I recommend to add some clarifications for non data scientists on the radiocarbon age re-calibration and the marine reservoir correction.

Researchers working with the database should not use it blindly and should already understand these common dating issues. The text already provides context and references for a deeper dive. "The radiocarbon ages in the database were recalibrated using the CALIB v8.1 with the IntCal20 (SHCal20), Marine20, or the mixed marine southern hemisphere radiocarbon calibration curve depending on the sample type and marine content (Reimer et al., 2009; Heaton et al., 2020). The source publications use different marine reservoir corrections, depending on the dated material, while our database standardizes the marine reservoir correction to 1144 ± 120 yr (Hall et al., 2010) for simplicity and consistency." The text was revised to improve clarity: "Converting measured radiocarbon activities to calendar age requires corrections for the variable atmospheric radiocarbon history, and for the reservoir age of the ocean."

l.355: Here, you could also provide an example of the second case, WDC and Byrd?

Revised as follows: "In the second case, ice temperatures remain cold at depth and reach a deeper englacial thermal minimum, which is marginally cooler than the surface ice, before they warm again towards the base such as at the site of the WAIS Divide Ice Core (WDC) and Byrd ice core …"

l.445: Always good to define the Holocene age period.

Addressed.

l.461/Fig.8: A median may provide a more intuitive metric of this long tail statistical distribution, although statistical mean and standard deviation may be consistently used in the error model.

The median value was added to the figure caption.

Sect. 3.1.: It could be helpful to follow the quality assessment in an example figure, or refer to already existing Figures (e.g. Fig. 5). Or maybe add a check table of criteria and sites for a better overview.

Revisions point the reader to Figure 3 to rectify this since it visually illustrates an example.

Table 1: Where does the factor 3 come from in the boreTemp data?

Arises from the fact that surface, basal and englacial temperatures are the most powerful constraints at each borehole profile. Table 1 was updated to minimize confusion and a sentence was added to Section 3.1.4: "Each respective borehole temperature profile is reduced to three datapoints (surface, englacial, and basal ice temperature) that most meaningfully represent the entire profile (Table 1)."

l.660 I very much support to add age structures as soon as possible (AntICE3), such an accurate ice age tracing module has been implemented in PISM recently.

This is the goal for the next iteration.

l.728: I understand, this is not a complete list of online data repositories, but Pangaea may also provide some data to be added (https://www.pangaea.de/?t=Cryosphere&f.location%5B%5D=Antarctica)

Agreed, we acknowledge that some data was missed and looking across additional data repositories would be valuable for future iterations of the database.

l.775: Update the to the final publication (part 2?): The Cryosphere, 14, 633–656, 2020, https://doi.org/10.5194/tc-14-633-2020

Corrected.

**data:**

It could be helpful to have a column (AntICEdat) that provides a quick check on which data have been updated with respect to Version 1 (as done for PaleoRSL).

Considering all the ages were recalibrated and site IDs were updated, identifying the data that was in the original AntICEdat database relative to the significantly expanded AntICE2 database does not add substantial value.

I would wish a consistent reordering of the most relevant data columns, as indicated above, for easier data
processing. The first 12 columns should include the calibrated age, while the raw data (e.g. uncorrected and corrected age) can be put further to the right of each sheet. The age uncertainty could be consistently defined as lower and upper bound (as for PaleoRSL), even when the uncertainty range is symmetric around the mean. Not all data types have a sample identifier SSID, which would be helpful to add..

See my response to comment 1. The paleoRSL dataset does specify if it is lower or upper proxy data by the "CTY"
or "Type" column. Some data types have SSID or SID because there many data points at a given site, while other data types do not, such as rdotGPS because there is only one observation per site. Difference between the datatype spreadsheets predominantly arise due to distinct differences in the proxy data in each datatype.

The PaleoRSL entry 9602 has a wrong sign in lon and lat.

Corrected.

Hat does a Tier of -1 would mean (for the consideration in model-data evaluation), as for instance paleoH site 1103?

A tier = -1 means you should exclude this observation based on the quality criteria described in Section 3.1. The text was revised with the following statement: "Some data are assigned to a tier of -1, which signifies that the observation should be entirely excluded for data-model comparison since it failed one or several quality criteria.
This tier equal to -1 category is solely for the purpose of logging the data and identifying it as not trustworthy to ensure exclusion from analyses."

**References not already mentioned in the article:**

Ely, Jeremy C., Chris D. Clark, David Small, and Richard CA Hindmarsh. "ATAT 1.1, the Automated Timing Accordance Tool for comparing ice-sheet model output with geochronological data." Geoscientific Mode
Development 12, no. 3 (2019, 933-953)

**Response to referee comments #2 (Steven Phipps)**

**General comments:**

The manuscript presents the AntICE2 database, an updated database of observational constraints on the historical evolution of the Antarctic Ice Sheet (AIS). The new database improves upon the previous version not only by including a much greater number of records, but also by classifying each record and explicitly quantifying the uncertainties.

Observational constraints are critical to constrain our understanding of the dynamics of the AIS, including the ice sheet models used to predict its future evolution. The authors correctly note that such constraints are currently lacking, particularly in the interior of the continent. The AntICE2 database is therefore an extremely valuable resource to the community, not just in order to improve our understanding of the past evolution of the AIS but also to improve future projections. The database is therefore of considerable scientific importance, and arguably even of broader socio-economic importance.
The authors are therefore to be commended for their efforts in constructing the AntICE2 database and in making it available to the community. I consider that the manuscript is clear, comprehensive and well written. I recommend it for publication, subject to consideration of the minor comments provided below.

We would like to thank Steven Phipps for his referee comments.

**Specific comments:**

Lines 29-31 and 737-740: Stronger statements could be made here, particularly in the Abstract. From the perspective of ice sheet modelling, databases of this nature provide much more than just accurate spin-ups.

Reliable reconstructions of past states of the AIS are also an essential part of the process of calibrating and evaluating ice sheet models. The authors could state this, as well as stating that the implementation of the AntICE2 database therefore has the potential to generate more accurate predictions of the future evolution of the AIS and, equally importantly, more accurate quantification of the uncertainties in those predictions.

We believe we effectively communicate this point in the abstract since paleo model spin ups to present day are broadly used to evaluate contemporary and future ice sheet changes. The abstract states, "… yield more robust paleo model spin ups for forecasting future ice sheet changes." which implies more accurate spin ups and ice sheet forecasting. The text was revised in Section 5 to emphasize this point: "This will lead to a more accurate understanding of contemporary and future changes of the AIS."

Line 59: The text could say "to evaluate and calibrate ice sheet models" (or similar) rather than just "for ice sheet models".

Text revised.

Line 150: Is there no reference or link of any description for the ICE-D database? Could the authors elaborate at least a little more on factors such as the contents, location and maintenance of this database, and whether/how members of the community might be able to access it?

A URL to the ICE-D database has been added to the manuscript (https://www.ice-d.org/). It is a well-established database which is regularly maintained by Greg Balco and his colleagues. They offer workshops on how to leverage the database for research. Technical publications on the database can be found through the provided URL.

Lines 165, 302, 303 and 364: Are the authors able to provide references or other justifications for the assumed
default uncertainties of 10m, 1m, 1m and 0.1 degrees Celsius respectively? Otherwise they seem to be excessively arbitrary.

These values are used in instances where no uncertainties are provided in the source literature. They are based on existing uncertainties found in other publications within the same datatype. Conservative uncertainty estimates are inferred that err on the side of caution. The text was revised to include the following: "Whenever information on uncertainties is lacking in the source publication, uncertainty estimates are judged conservatively by relevant expert members of the author team or are derived from other studies using the same datatype."

Section 3.2: The depth-age relationships derived from individual ice cores still provide a valuable constraint for ice sheet models, even in the absence of radiostratigraphy or any other form of spatial extrapolation. Hence I would strongly encourage the authors to consider including this data in future versions of the database.

Agreed, the text was revised as such: "The depth-age data from ice cores can directly constrain the age structure. Moreover, there are some regional age reconstructions for …"

Technical corrections
  • Line 184: Add a comma after "local".
Corrected.
  • Line 223: Replace "Sometimes resulting" with "Sometimes this results".
Sentence rephrased.
  • Line 234: Add a comma after "rich".
Corrected.
  • Line 400: The word "associated" might be better than "affiliated".
Corrected.
  • Line 438: Replace "On the modeller side" with "From the perspective of modellers" (or similar).
Corrected.
  • Line 495: Replace "most constraining power on" with "greatest power to constrain".
Corrected.
  • Line 631: Remove the word "internally".
Corrected.
  • Line 645: Replace "being" with "represent".
Corrected.